# Unveiling the Multifaceted Role of HP6: A Critical Regulator of Humoral Immunity in *Antheraea pernyi* (Lepidoptera: Saturniidae)

**DOI:** 10.3390/ijms26104514

**Published:** 2025-05-09

**Authors:** Chengbao Liu, Jinzhu Yu, Ting Fu, Xueshan He, Lin Zhao, Xialu Wang, Rong Zhang

**Affiliations:** 1School of Life Science and Bio-Pharmaceutics, Shenyang Pharmaceutical University, Shenyang 110016, China; liuchengbao7012@163.com (C.L.); yu13030731705@163.com (J.Y.); ahexueshan7432@163.com (X.H.); alina1956@163.com (L.Z.); 2School of Medical Devices, Shenyang Pharmaceutical University, Shenyang 110016, China

**Keywords:** serine protease, *Antheraea pernyi*, humoral immune, PPO-activation system, AMP synthesis

## Abstract

Serine proteases are widely distributed in both invertebrates and vertebrates, playing critical roles in the regulation of innate immunity. In the insect innate immune system, two pivotal pathways—the prophenoloxidase (PPO) activation cascade and Toll pathway-mediated antimicrobial peptide (AMP) synthesis—are both tightly regulated by serine protease cascades. This study focuses on serine protease–hemolymph protease 6 of *A. pernyi* (*Ap*-HP6). Following immune stimulation, the expression of *Ap*-proHP6 was significantly induced, primarily observed in hemocytes and the fat body. After suppressing *Ap*-proHP6 expression via RNA interference (RNAi) and infecting larvae with different microbes, the expression levels of AMPs showed a downward trend. When endogenous *Ap*-proHP6 content in hemolymph was reduced using RNAi technology or anti-r*Ap*-proHP6-His_6_ polyclonal antibodies, PAMPs/microbe-mediated phenoloxidase (PO) activity significantly decreased. These results suggest that *Ap*-HP6 has a positive regulatory effect on PPO activation and AMP synthesis. Additionally, the in vitro hydrolysis of r*Ap*-proHP6-Tb-His_6_ yielded r*Ap*-HP6 with serine protease activity, which exhibited optimal reaction conditions for S-2288 at pH 8.0, 50 °C, and 15 min.

## 1. Introduction

*A. pernyi* is a species of wild silk insect belonging to the order *Lepidoptera* in China, with a long history and cultural significance. In addition to serving as a raw material for the textile industry, *A. pernyi* products at various life stages—including the eggs, larvae, cocoons, and adults—hold significant medicinal value. For instance, *A. pernyi* is commonly used in traditional medicine to treat conditions such as atherosclerosis and non-alcoholic fatty liver disease (NAFLD) [1]. Additionally, *Cordyceps militaris* plays a crucial role in tonifying the lungs and kidneys, and inducing cancer cells apoptosis [2,3]. However, as *A. pernyi* is predominantly raised in wild environments, it faces challenges, which severely impact its life cycle and yield. Applying modern molecular biology techniques to identify functional proteins in *A. pernyi* and investigate its immune defense mechanisms could enhance its edible value and increase its production.

Innate immunity serves as the first line of defense in an organism’s immune system and has remained highly conserved throughout evolution, from insects to mammals [4]. However, insects, the most diverse group of organisms on Earth, have developed a variety of strategies to combat microbial invasion [5]. Due to their lack of adaptive immunity—such as antibodies and immune memory—models like *Drosophila* have been instrumental in advancing the research on innate immune responses [6]. Studies on fruit flies have uncovered the Toll and immune deficiency (IMD) signaling pathways have been identified as critical components of insect innate immunity [7,8]. Interestingly, the Toll pathway closely parallels the mammalian interleukin-1 receptor (IL-1R) and myeloid differentiation primary response 88 (MyD88)-dependent Toll-like receptor (TLR) signaling pathways, providing valuable insights into human immune signaling mechanisms [9]. Insect innate immunity comprises both humoral and cellular responses. The humoral immune response is a highly efficient defense mechanism that is rapidly activated upon pathogen recognition by pattern recognition receptors (PRRs), which detect pathogen-associated molecular patterns (PAMPs) [10]. During bacterial invasion, insects induce the synthesis of AMPs and other antimicrobial compounds via signaling pathways, such as Toll, IMD, and JAK-STAT. On the other hand, it activates the PPO activation system (PPO-AS), leading to the production of active PO, which catalyzes melanin formation and ultimately entraps and eliminates invading pathogens [11].

In humoral immune signaling pathways, the serine protease cascade serves as a key regulator and activator of hemolymph coagulation, melanization, and AMPs production. A series of serine proteases become sequentially activated, amplifying the original signal to facilitate a rapid and efficient response to microbial infections [12]. Serine proteases are also essential in mammalian physiological processes. The complement system defends against pathogens through rapid serine protease activation, generating biologically active products [13]. As proteolytic enzymes, serine proteases have been extensively studied for their structural and functional properties. Their typical structure consists of two β-sheet aggregates arranged in a globular conformation. The active site residues His^57^, Asp^102^, and Ser^195^ (encoded by *chymotrypsin*) are located between the two aggregates [14]. In vivo, inactive serine proteases exist as zymogens without enzymatic activity. Upon activation, the Clip domain is hydrolyzed and covalently linked to the C-terminal serine protease domain via disulfide bonds, ensuring structural stability and functional activity [15]. When mutations or deletions occur in the catalytic triad, these enzymes lose their proteolytic activity and are termed serine protease homologs (SPHs) [16].

To date, numerous genes encoding serine proteases and their homologs have been identified in insects. For example, *Bombyx mori* possesses 143 serine protease (SP) and SPH genes, *Drosophila melanogaster* has 257, *Helicoverpa armigera* has 46, *Aedes aegypti* has 63, *Manduca sexta* has 125, and *Tenebrio molitor* has 200 [17,18,19,20,21,22]. In *T. molitor*, the serine protease cascade amplification system involves peptidoglycan recognition protein-SA (PGRP-SA), which recognizes lysine-type peptidoglycan in the presence of Gram-negative binding protein 1 (GNBP-1) [23]. This initiates the modular serine protease (MSP)–serine protease AE (SAE)–spätzle-processing enzyme (SPE) cascade, leading to the activation of proSPH1, PPO, and pro-spätzle, which in turn triggers melanization and AMP production [24,25,26]. In *M. sexta*, the recognition of invading pathogens by PRRs activates a complex extracellular SP-SPH cascade involving HP14, HP21, proHP1, HP6, HP8, PAP1-3, SPH1, and SPH2, which mediate melanin formation and AMP synthesis [27,28]. Currently, four PPO cascade activation pathways have been identified: HP14-HP21-PAP2/3, proHP1*-HP6-PAP1, HP14-HP2-PAP2, and HP14-HP21-HP5-HP6-PAP1 [29,30,31,32,33]. Genetic studies in *D. melanogaster* have elucidated the Toll activation pathway, comprising the initiating MSP, the terminal SPE, and the clip protease Grass, which connects them [6,24,34,35]. An alternative spätzle activation pathway involves the clip protease Persephone, independent of MSP [36,37]. The *D. melanogaster* melanization cascade involves the clip proteases MP1, MP2, and Hayan, with MP2 functioning as the PPO-activating protease for *Drosophila* PPO1 [37,38]. However, the precise protease cascade linking pro-spätzle activation and PPO in *D. melanogaster*’s immune response remains unclear.

*A. pernyi* serves as an ideal model for the innate immunity research—particularly humoral immunity—due to its sequenced genome, abundant larval hemolymph, and highly sensitive immune defense mechanisms. This study aimed to explore the humoral immune function of *Ap*-HP6 in *A. pernyi*. Furthermore, the enzymatic activity of recombinant *Ap*-HP6 (r*Ap*-HP6) was analyzed to determine its optimal catalytic conditions, laying the groundwork for future applications of *Ap*-HP6 in medicine and functional foods.

## 2. Results

### 2.1. Ap-HP6 Is Involved in the Immune Response of A. pernyi

To identify immune-related proteins in *A. pernyi* larvae, RNA sequencing was conducted to analyze DEGs in second-instar larvae stimulated by pathogenic bacteria. The results reveal that 2593 genes are upregulated, whereas 1161 genes are downregulated after 9 h of immune induction (Appendix A). GO functional annotation showed that DEGs were primarily associated with molecular function, cellular components, and cellular processes. In terms of the molecular function, DEGs were significantly enriched in pathways related to “tRNA dihydrouridine synthase activity”, “serine hydrolase activity”, “scavenger receptor activity”, “oxidoreductase activity”, and “enzyme regulator activity” (Figure 1A). Notably, genes associated with serine hydrolase and oxidoreductase activity were the most prevalent. In insects, serine proteases are key components of the serine protease cascade in the humoral immune response. To identify serine hydrolases involved in the *A. pernyi* serine protease cascade, genes with significantly upregulated expression and serine hydrolase activity following immune stimulation were further screened based on statistical significance (*p* < 0.001) and FC ≥ 4. As shown in Figure 1B, 33 genes encoding putative serine hydrolases were identified. Here, we focus on the serine protease cascade of immune factors related to the humoral immunity of *A. pernyi*. However, among these, only *Ap*-proHP6 had been previously reported at the mRNA level. Therefore, this study focused on *Ap*-HP6 to determine whether it participates in the larval humoral immune response via the serine protease cascade.

To further investigate the biological functions of *Ap*-HP6 (particularly its immune functions), this study conducted a bioinformatic structural analysis of *Ap*-proHP6 (GenBank ID: ANJ42865.1). A phylogenetic tree was constructed using 89 serine proteases known to be involved in immune responses across representative species, including microorganisms, insects, marine organisms, zebrafish, and mammals (Figure 1C). Despite their phylogenetic divergence, *Ap*-proHP6 clustered closely with eight serine proteases from *D. melanogaster*, *M. sexta*, and *A. gambiae*, demonstrating a high degree of evolutionary conservation. Protein structure plays a critical role in determining function. To predict the potential immune function of *Ap*-HP6, its spatial structure was compared with the eight highly homologous immune-related serine proteases mentioned above. As shown in Figure 1D, the carboxy-terminal serine protease domains of all nine serine proteases exhibited significant structural conservation. The catalytic triad, consisting of His^158^, Asp^208^, and Ser^307^ (encoded by *Ap*-proHP6), was highly conserved and located within the sequences ^155^TAAHC^159^, ^208^DIAL^211^, and ^305^GDSGGP^310^, respectively. Furthermore, three disulfide bonds reinforced the catalytic triad, enhancing structural stability and enzymatic efficiency. Several members of the serine protease family, including *Ms*HP2, *Ms*HP6, *Ms*HP21, *Ag*SPC9, *Dm*Snake, *Dm*Spirit, *Dm*Hayan, and *Dm*Psh, are known to be critical components of the humoral serine protease cascade. Based on structural analysis, it is hypothesized that *Ap*-HP6 functions as an immune-related serine protease in *A. pernyi*. It is likely involved in the serine protease cascade, regulating the activation of the PPO-AS and Toll pathway-mediated synthesis of AMPs. To further characterize Ap-HP6, the major domains of the proenzyme *Ap*-proHP6 and its active enzyme form *Ap*-HP6 were predicted. As shown in Figure 1E, *Ap*-proHP6 belongs to the CLIP subfamily, with an N-terminal clip domain, followed by a linker region and a C-terminal serine protease domain characteristic of the trypsin family. A disulfide bond (^99^S-S^228^) connects the linker region to the trypsin domain. Upon specific proteolytic cleavage at position ^113^Y-I^114^, *Ap*-proHP6 is activated, and despite cleavage, the clip and trypsin domains remain covalently linked to form the active enzyme *Ap*-HP6.

For functional studies, an *E. coli* expression system was utilized to produce r*Ap*-proHP6-His_6_, which was successfully purified using Ni-Sepharose chromatography columns. SDS-PAGE analysis revealed that r*Ap*-proHP6-His_6_ migrated as a single band of approximately 38 kDa, consistent with the predicted molecular weight. The purified recombinant protein was subsequently used as an antigen to generate a polyclonal rabbit antibody against r*Ap*-proHP6-His_6_, achieving a titer of 1:64,000 (Figure 2C). Western blot analysis confirmed that both r*Ap*-proHP6-His_6_ and native *Ap*-proHP6 appeared as single bands, indicating the specificity of the antibody. These results provide a crucial foundation for further investigations into the immune function of *Ap*-HP6

### 2.2. Expression Profiles of Ap-HP6 in Response to Immune Challenge

To explore the potential role of *Ap*-HP6 in the immune response of *A. pernyi*, the *Ap*-proHP6 mRNA expression profile was analyzed following different microbial injections using RT-qPCR. The expression of *Ap*-proHP6 mRNA in second-instar larvae was first assessed after microbial stimulation.

As shown in Figure 2A, upon injection with *S. aureus* and *C. albicans*, *Ap*-proHP6 mRNA levels increased slightly within the first 3 h, followed by a significant upregulation over the next 3 h. The transcript levels peaked at 6 h post-injection, then rapidly declined within the next 3 h, maintaining a relatively low induced expression until 36 h, eventually returning to baseline at 48 h. In contrast, following *E. coli* injection, *Ap*-proHP6 mRNA levels exhibited a modest elevation within the first 6 h, followed by a sustained increase, peaking at 9 h. The expression then sharply declined over the next 3 h and returned to baseline by 48 h post-injection. Additionally, *Ap*-proHP6 expression was assessed in different tissues 6 h post-injection. Under normal physiological conditions, *Ap*-proHP6 mRNA was detected in various tissues, including the fat body, epidermis, midgut, Malpighian tubules, and hemocytes, with hemocytes identified as the primary site of expression. Upon microbial injection, *Ap*-proHP6 mRNA levels significantly increased in the fat body and hemocytes compared to other tissues (Figure 2B). A Western blot analysis further evaluated *Ap*-proHP6 protein levels in the hemolymph following microbial challenge. As shown in Figure 2C, *Ap*-proHP6 protein expression was significantly upregulated in response to all three microorganisms, though with varying expression levels. After *C. albicans* injection, *Ap*-proHP6 protein expression initially declined (~0.7-fold) within the first 6 h, followed by continuous upregulation, peaking at 24 h (~1.5-fold). In *S. aureus*-infected larvae, *Ap*-proHP6 protein levels peaked at 6 h (~1.4-fold), then rapidly declined over the next 6 h and returned to baseline at 36 h. Following *E. coli* injection, *Ap*-proHP6 protein expression was significantly upregulated, peaking at 9 h (~2.6-fold) and sustaining elevated levels for 15 h before returning to baseline within 48 h. Notably, an unidentified protein band was observed above *Ap*-proHP6 between 12 and 72 h post-*E. coli* injection. These findings suggest that *Ap*-HP6 is an immune-related protein that likely plays a crucial role in *A. pernyi* immune defense against microbial infections.

### 2.3. Involvement of Ap-HP6 in AMPs Production

Although *Ap*-HP6 has been identified as an immune-related protein, its precise role in the *A. pernyi* immune system remains unclear. To determine its contribution to AMP synthesis, *Ap*-proHP6 mRNA was knocked down using RNAi, and its efficiency was assessed using RT-qPCR. As shown in Figure 3A, *Ap*-proHP6 mRNA levels in ds*Ap-proHP6*-treated larvae were reduced by over 90% at 48 h post-injection. Western blot analysis confirmed this knockdown at the protein level, with *Ap*-proHP6 expression in the hemolymph decreasing by approximately 70% compared to ds*EGFP*-treated control larvae. To investigate the role of *Ap*-HP6 in AMP induction, RNAi-treated larvae were challenged with microbial infections. As shown in Figure 3B, *C. albicans* injection led to a marked downregulation of AMP gene expression in *A. pernyi* larvae. However, after *Ap*-proHP6 knockdown followed by *S. aureus* injection, most AMP transcript levels significantly decreased, except for lebocin, which was notably upregulated. Similarly, following *E. coli* injection, attacin, lebocin, and moricin transcript levels declined, while defensin and lysozyme mRNA levels increased. Notably, the knockdown of *Ap*-proHP6 expression led to a concomitant decrease in spätzle transcript levels in both *S. aureus*- and *C. albicans*-challenged samples, indicating that *Ap*-proHP6 potentially regulates AMP synthesis through the Toll signaling pathway (Appendix A). These results suggest that *Ap*-HP6 may play a positive regulatory role in AMP synthesis.

### 2.4. Contribution of Ap-HP6 to the PPO Activation System in A. pernyi

To determine the role of *Ap*-HP6 in the PPO-AS, *Ap*-proHP6-deficient plasma was prepared using RNAi, and in vitro PO activity assays were performed. Compared to untreated or ds*EGFP*-treated plasma, a significant reduction in PO activity was observed in *Ap*-proHP6-knockdown plasma following elicitor stimulation (Figure 4A). Since ds*EGFP*-treated plasma exhibited no significant difference from native plasma, the observed decrease in PO activity upon *Ap*-proHP6 knockdown suggests that endogenous *Ap*-HP6 positively regulates PPO activation. To further confirm the role of *Ap*-HP6 in PPO-AS, purified anti-r*Ap*-proHP6-His_6_ antibodies were added to plasma to assess the impact of endogenous *Ap*-proHP6 depletion at the protein level. As shown in Figure 4B, elicitor addition significantly increased PO activity in plasma, whereas the presence of the antibody led to a marked reduction in PO activity compared to untreated control plasma. In addition, using RNAi technology, we prepared *Ap*-proHP6-deficient fifth-instar larvae and stimulated them with a mixed bacterial solution. In Appendix A, compared to the naive control group, the number of melanized nodules on the cuticle of fifth-instar *A. pernyi* larvae increased significantly following stimulation with the mixed microbial suspension. After microbial challenge, the extent of cuticular darkening was similar between the naive control group and the ds*EGFP*-treated group. In contrast, larvae with *Ap*-proHP6 knockdown exhibited a marked reduction in melanized nodules and less-pronounced cuticular darkening compared to the two control groups. These results demonstrate that *Ap*-HP6 has a positive regulatory effect on PPO-AS, and the expression of *Ap*-HP6 can induce the melanization response.

### 2.5. Serine Protease Enzymatic Activity Assay of rAp-HP6

The evidence that endogenous *Ap*-HP6 is involved in PPO activation in *A. pernyi* hemolymph (Figure 4) and AMP synthesis via the Toll signaling pathway (Figure 3 and Appendix A) suggests that *Ap*-HP6 may function as a component of the serine protease cascade system in *A. pernyi*. To further investigate its role in serine protease cascades, PPO activation, and AMP synthesis, r*Ap*-proHP6-Tb-His_6_ was generated by inserting a thrombin cleavage site (^114^LVPRGS^119^) at the predicted zymogen activation site. The recombinant protein was then cleaved in vitro with thrombin to produce r*Ap*-HP6 with serine protease activity, and its enzymatic properties were analyzed. r*Ap*-proHP6-Tb-His_6_ was incubated with thrombin at various molar ratios for different durations, and the extent of proteolysis under varying catalytic conditions was examined via SDS-PAGE. Densitometric analysis of hydrolyzed protein bands determined that the optimal conditions for the thrombin-mediated cleavage of r*Ap*-proHP6-Tb-His_6_ were a thrombin to r*Ap*-proHP6-Tb-His_6_ molar ratio of 1:70, incubated at 23 °C for 6 h (Appendix A). Under these conditions, r*Ap*-proHP6-Tb-His_6_ was predominantly cleaved into two protein bands of 26.7 and 11.8 kDa (Figure 5A, Lane 5, arrows), closely matching the predicted molecular weights of the two subunits generated from *Ap*-proHP6 cleavage. In contrast, r*Ap*-proHP6-His_6_ lacking the thrombin cleavage site was also cleaved, yielding bands of 24.6 and 12.9 kDa, which did not correspond to the expected proHP6 activation site. This discrepancy may be due to non-specific hydrolysis at the “^234^LAPRA^238^” sequence present in *Ap*-proHP6. Additionally, non-reducing SDS-PAGE analysis revealed that, despite thrombin cleavage, r*Ap*-proHP6-Tb-His_6_ remained as a single 35 kDa band (Appendix A), suggesting that the 26.7 and 11.8 kDa subunits were linked by an intermolecular disulfide bond. This observation aligns with the previous reports that active serine proteases often retain disulfide linkages between subunits following zymogen activation. These findings suggest that r*Ap*-proHP6-Tb-His_6_, when cleaved in vitro by thrombin, may form an enzymatically active r*Ap*-HP6.

To further evaluate the catalytic activity of r*Ap*-HP6 as a serine protease, its ability to hydrolyze the serine protease substrate S-2288 was examined. As shown in Figure 5B, neither r*Ap*-proHP6-His_6_ nor r*Ap*-proHP6-Tb-His_6_ alone exhibits serine protease activity (columns 3 and 5). However, both proteins were able to catalyze the chromogenic reaction of S-2288 following thrombin cleavage, with the cleavage product of r*Ap*-proHP6-Tb-His_6_ (i.e., putative r*Ap*-HP6) displaying the highest catalytic activity (column 6). Additionally, thrombin, being a serine protease, exhibited some hydrolytic activity against S-2288. However, the catalytic activity of the putative r*Ap*-HP6 was significantly higher than the combined activities of r*Ap*-proHP6-Tb-His_6_ and thrombin alone (column 6 > column 2 + column 5). Following confirmation that hydrolyzed r*Ap*-HP6 exhibited serine protease activity, its optimal reaction conditions were determined to be pH 8.0, 50 °C, and an incubation time of 15 min (Figure 5C–E). Under these conditions, the maximum reaction velocity (V_max_) was measured at 1.325 U/min, with a Michaelis constant (K_m_) of 380.0 nM (Figure 5F).

### 2.6. The Effect of Exogenous rAp-HP6 on PP0-AS

As previously demonstrated, endogenous *Ap*-HP6 participates in the activation of hemolymph PPO in *A. pernyi* in response to pathogen stimulation. After confirming that r*Ap*-proHP6-Tb-His_6_ can be hydrolyzed to generate r*Ap*-HP6 with serine protease activity, we further investigated the effect of exogenous r*Ap*-HP6 on the PPO-AS in the *A. pernyi* hemolymph. As shown in Figure 6, in the absence of stimulants, the addition of buffer (column 1), thrombin (column 2), r*Ap*-proHP6-His_6_ (column 3), or r*Ap*-proHP6-Tb-His_6_ (column 4) to *A. pernyi* hemolymph does not induce any significant changes in PO activity. However, the introduction of exogenously hydrolyzed r*Ap*-HP6 results in a slight increase in natural hemolymph PO activity (column 5). In contrast, when PPO is significantly activated by PAMPs or microbial stimulation (column 6), the addition of thrombin alone has no impact on this activation (column 7), whereas r*Ap*-proHP6-His_6_ and r*Ap*-proHP6-Tb-His_6_ exhibit a mild promoting effect (columns 8 and 9). This effect may be attributed to the presence of an upstream serine protease induced by the stimulant, which hydrolyzes *Ap*-proHP6 in the hemolymph PPO-AS. Notably, the addition of the *Ap*-proHP6-Tb-His_6_+Tb mixture significantly enhances PAMP/microbe-mediated PPO activation (column 10), suggesting that hydrolyzed r*Ap*-HP6, the active form of exogenous HP6, actively participates in pathogen-induced PPO-AS activation in the *A. pernyi* hemolymph.

## 3. Discussion

*A. pernyi* is an economically important insect in China, holding significant value not only in the textile industry but also in functional food and pharmaceutical applications. During immune responses, *A. pernyi* produces diverse bioactive substances, including AMPs, serine proteases, and POs, which exhibit antibacterial, antiviral, and antitumor activities, providing critical resources for novel drug development. Serine proteases are widely present in both invertebrates and vertebrates. Beyond their role in regulating innate immunity, they play essential roles in various physiological and pathological processes. However, the immune-related functions of serine proteases in *A. pernyi* remain underexplored. This study identifies, for the first time, a humoral immune response-associated serine protease—*Ap*-HP6—in *A. pernyi*. Through systematic experiments, we demonstrate that *Ap*-HP6 is a key component of humoral immunity, participating in the activation of PPO and the production of AMPs (the experimental methodology flowchart is presented in Appendix A).

To comprehensively explore immune-related serine proteases in *A. pernyi*, we conducted comparative transcriptome sequencing analysis of the insect before and after pathogen stimulation. The sequencing results revealed 2593 upregulated genes, from which 33 serine protease-like proteins were screened (Figure 1B and Appendix A). Among these, serine protease cascade-related immune factors potentially involved in the humoral immunity of insects include: serine proteinase-like protein 1 [*Helicoverpa armigera*], serine proteinase-like protein 2 [*Manduca sexta*], serine protease snake [*Papilio xuthus*], serine protease HP21 precursor [*Bombyx mori*], hemolymph proteinase 6 [*Antheraea pernyi*], hemolymph proteinase 7/9 [*Manduca sexta*], hemolymph protein 14 [*Bombyx mori*], serine proteinase [*Samia ricini*], serine protease 5, partial [*Lonomia obliqua*], prophenoloxidase activating enzyme precursor [*Manduca sexta*], prophenoloxidase-activating proteinase-3 precursor [*Manduca sexta*], prophenoloxidase activating factor 1, partial [*Lonomia obliqua*], prophenoloxidase activating factor 3 [*Bombyx mori*], and prophenoloxidase-activating proteinase [*Samia ricini*]. The prophenoloxidase activating enzyme precursor [*Manduca sexta*], prophenoloxidase-activating proteinase-3 precursor [*Manduca sexta*], and prophenoloxidase-activating proteinase [*Samia ricini*] serve as terminal effector proteins of the PPO-AS and have been extensively characterized. Additionally, serine proteinase-like protein 1 [*Helicoverpa armigera*] and serine proteinase-like protein 2 [*Manduca sexta*], as serine protease homologs, lack the catalytic triad and exhibit no enzymatic activity; their mechanisms in immune regulation require further investigation. In contrast, hemolymph proteinase 6 [*Antheraea pernyi*], hemolymph proteinase 7/9 [*Manduca sexta*], and hemolymph protein 14 [*Bombyx mori*], as canonical serine proteases, have drawn significant research interest. However, only the full-length cDNA sequence of *A. pernyi* proHP6 has been reported to date, while complete sequences for proHP7/9/14 remain uncharacterized. Therefore, this study selected *A.p*-HP6 as the research target to investigate its immune-related functions. As shown in Figure 1C,D, the amino acid sequence of *Ap*-proHP6 exhibits 54% identity with *Ms*-proHP6. As outlined in the Introduction Section, *Ms*-proHP6 is definitively involved in two PPO activation pathways in *M. sexta*: proHP1*-HP6-PAP1 and HP14-HP21-HP5-HP6-PAP1 cascades. Additionally, it participates in AMP synthesis [31,32]. Thus, the research on proHP6 in *M. sexta* innate immunity provides significant insights for our study. Moreover, it has been reported in the literature that *Ms*-proHP2, *Ms*-proHP6, *Ms*-proHP21, *Ag*-proSPC9, *Dm*Snake, *Dm*Spiri, *Dm*Hayan, and *Dm*Psh, which are highly conserved in structure with *Ap*-proHP6, are all major members of the serine protease cascade system associated with humoral immune responses in each species [15,31,37,38]. Given that structural homology often implies functional similarity, we hypothesize that *Ap*-HP6 likely participates in humoral immune processes.

To investigate the role of *Ap*-HP6 in humoral immune responses, we first detected the tissue distribution of *Ap*-proHP6 expression and the changes in its mRNA expression at different times following pathogen stimulation. Under basal conditions, *Ap*-proHP6 was primarily expressed in the fat body and hemocytes, with baseline expression also detected in the midgut, epidermis, and Malpighian tubules. After immune stimulation, *Ap*-proHP6 expression was significantly upregulated in all these tissues, with the most pronounced increases observed in the fat body and hemocytes (Figure 2A,B). It is well-established that the insect fat body and hemocytes are functionally analogous to the liver and white blood cells in mammals, respectively. These tissues synthesize and secrete various immune-related proteins into the hemolymph, playing pivotal roles in regulating humoral immune responses. Similarly, in *A. gambiae*, immune stimulation enhances the transcriptional abundance of the serine protease *Ag*-Sp22D in the midgut epithelium, fat body, and hemocytes [39]. In *M. sexta*, immune challenge induces a 2.3-fold increase in *Ms*-HP1b mRNA levels in hemocytes [40]. Both serine proteases have been shown to critically contribute to humoral immune responses. On the other hand, the expression of *Ap*-proHP6 varied significantly following stimulation with different pathogens. Compared to *E. coli* and *S. aureus*, *Ap*-proHP6 mRNA levels peaked at the highest magnitude 6 h post-*C. albicans* challenge. In contrast, after *E. coli* injection, *Ap*-proHP6 mRNA expression reached its peak at 9 h (Figure 1D). Similarly, in *A. gambiae*, the immune response involving CLIPA7 shows rapid upregulation (detectable in 1–3 h) for *C. albicans* and *S. aureus*, while the response to *E. coli* is delayed [41]. This pattern is also observed in *Aedes aegypti* (*Ae*-CLIPB15 and *Ae*-CLIPB22) and *Aedes albopictus* (*Aa*Atg8) [42,43]. We hypothesize that *C. albicans* infection activates distinct immune signaling pathways, such as the CLRs-serine/threonine kinase pathway, which may more efficiently promote serine protease gene expression [44]. Notably, *C. albicans* (a fungus) possesses a cell wall rich in chitin and mannan, whereas *S. aureus* (a Gram-positive bacterium) has a peptidoglycan-dominated cell wall. Upon infection, insect cells rapidly recognize PAMPs via PRRs, activating immune pathways, like the Toll and Imd pathways. In contrast, *E. coli* (a Gram-negative bacterium) contains LPS in its cell wall, which triggers the LPS-TLRs-NF-κB pathway—a more complex cascade involving intracellular phosphorylation and transcription factor activation [45]. This likely explains the slower response to *E. coli* compared to *C. albicans* and *S. aureus*. These results suggest that *Ap*-HP6 may play multifaceted roles in fungus-induced immune responses, leading to its rapid and robust upregulation. Additionally, *Ap*-HP6 could not only promote sensitive and rapid humoral immune responses (e.g., PPO activation and AMP synthesis) but also participate in cellular immunity, such as phagocytosis, encapsulation, and nodule formation.

AMPs are the most critical effector molecules in insect humoral immunity, synthesized via the IMD and Toll signaling pathways. AMPs typically appear in the hemolymph 6–12 h post-infection and directly kill pathogens by disrupting their membranes or interfering with internal mechanisms. As illustrated in Figure 3B, the knockdown of *Ap*-proHP6 significantly reduced the transcript levels of attacin and moricin in response to induction by all three microorganisms. However, defensin and lysozyme mRNA levels markedly increased after *E. coli* induction, while lebocin transcription was significantly upregulated upon *S. aureus* stimulation. Previous studies report that injecting protein *Ms*-proHP5 in *M. sexta* dramatically elevated attacin and moricin mRNA levels [31]. Additionally, RNA-seq analyses of *Aedes aegypti* and *Anopheles gambiae* under microbial induction revealed that bacterial infection responses primarily involve an upregulated expression of defensins and lysozyme [46]. Although *Ap*-HP6 modulates the expression of multiple AMPs, the specific signaling pathways mediating each AMP’s expression remain unclear. However, *Ap*-HP6 is definitively linked to Toll pathway-mediated AMP synthesis, as the RT-qPCR data show that *Ap*-proHP6 knockdown significantly reduces spätzle mRNA levels induced by Gram-positive bacteria and fungi (Appendix A).

The insect melanization response is a critical innate immune mechanism mediated by serine protease cascades that activate PPO. This process plays a vital role in defending against microbial infections. In this study, we investigated the regulatory role of *Ap*-HP6 in the PPO-AS of *A. pernyi* through three approaches: (1) In vivo RNAi-mediated knockdown of the *Ap*-proHP6 gene; (2) Neutralization of endogenous *Ap*-proHP6 protein using specific antibodies; and (3) Suppression of *Ap*-proHP6 expression to assess its impact on pathogen-induced melanization in fifth-instar larvae (Figure 4 and Appendix A). The results consistently demonstrate that *Ap*-HP6 positively regulates PPO-AS, primarily by modulating immune defenses against pathogens. Notably, under identical concentrations, the hemolymph of naive *A. pernyi* larvae exhibited differential sensitivity to PPO activation when stimulated by six microbial agents. Furthermore, replacing microbial activators with soluble PAMPs altered the sensitivity of hemolymph to PPO activation. One explanation for this is that *A. pernyi* hemolymph shows different sensitivities to different PAMPs within a specific range of PO activity stimulation (10–25 U). This disparity may reflect evolutionary adaptations to pathogens commonly encountered by *A. pernyi* in its natural habitat. Given the diversity and abundance of PAMPs on microbial surfaces, the observed PO activity levels (Figure 4A) represent the combined effects of all surface PAMPs, explaining the differential responses to microbial stimuli [47]. Additionally, we examined how *Ap*-HP6 suppression affects pathogen-induced melanization on the cuticle of fifth-instar larvae. The results show that larvae with *Ap*-proHP6 knockdown exhibit prominent melanotic nodules and less-noticeable cuticular darkening, indicating an enhanced melanization response to pathogen invasion (Appendix A). Previous studies have reported that, in *B. mori*, *Bm*SPH-1 RNAi resulted in reduced PO activity in hemolymph after *E. coli* challenge, while decreased *Bm*SPH-1 mRNA levels during the spinning or prepupal stages disrupted pupation and pupal cuticle melanization [48]. Although the research on the serine protease cascade system in *A. pernyi* remains limited, our findings align with reported functions of serine proteases in other insects. Specifically, *Ap*-HP6 acts as a component of the serine protease amplification cascade, participating in PPO activation and cuticular melanization when *A. pernyi* is exposed to external stimuli or infections.

Based on the above findings, *Ap*-HP6 is likely a key component of the serine protease cascade amplification system, regulating the synthesis of AMPs and PPO-AS in *A. pernyi* through its enzymatic activity. To elucidate the mechanistic role of *Ap*-HP6 in the serine protease cascade, we further investigated the proteolytic activity of *Ap*-HP6. Using recombinant protein engineering, we generated r*Ap*-proHP6-Tb-His_6_ and hydrolyzed it in vitro to produce r*Ap*-HP6. The zymogen r*Ap*-proHP6-Tb-His_6_ was hydrolyzed by thrombin at the cleavage site ^113^Y-I^114^, generating two fragments: an 11.8 kDa clip domain and a 26.7 kDa trypsin-like domain (Figure 5A). Non-reducing SDS-PAGE analysis revealed that thrombin-cleaved r*Ap*-proHP6-Tb-His_6_ migrated as a single band at ~35 kDa, confirming the formation of intramolecular disulfide bonds between the two fragments (Appendix A). To assess whether the cleaved product (r*Ap*-HP6) retains serine protease activity, we performed enzymatic assays using the chromogenic substrate S-2288 (D-Ile-Pro-Arg-p-nitroaniline). The results demonstrate that r*Ap*-HP6 exhibits robust serine protease activity (Figure 5B). S-2288, a broad-spectrum and highly sensitive chromogenic substrate for serine proteases, is hydrolyzed via nucleophilic attack by the catalytic serine hydroxyl group on the peptide bond’s carbonyl carbon, forming a tetrahedral transition state. Subsequent peptide bond cleavage releases p-nitroaniline (pNA), which absorbs maximally at 405 nm [49]. Finally, as shown in Figure 6, r*Ap*-HP6 generated in vitro and the activated HP6 produced in vivo through the serine protease cascade both exhibit positive regulatory effects on PPO activation. These findings not only elucidate the structural activation mechanism of HP6, but also establish r*Ap*-proHP6-Tb-His_6_—a recombinant protein with enzymatic activity identical to native *Ap*-proHP6—as a critical reagent for further studies on its potential antimicrobial and antitumor functions.

## 4. Materials and Methods

### 4.1. Insects, Microorganisms, and PAMPs

Second and fifth-instar *A. pernyi* larvae were obtained from Shenyang Agricultural University and reared on fresh *A. pernyi* leaves under laboratory conditions at 25 ± 1 °C. Bacterial and fungal strains used in the experiments were donated by another laboratory, including the fungi *Candida albicans* (CMCC98001), Gram-positive *Staphylococcus aureus* (CMCC26003), and Gram-negative *Escherichia coli* (CMCC44102). Pathogen-associated molecular patterns (PAMPs) used in this study included lipopolysaccharide (LPS; from *E. coli* O55:B5, L2880), laminarin (from *Laminaria digitata*, L9634), lipoteichoic acid (LTA; from *B. subtilis*, L3265), lysine-type peptidoglycan (Lys-PGN; from *Micrococcus luteus*, 53243), diaminopimelic acid-type peptidoglycan (DAP-PGN; from *B. subtilis*, 69554), and mannan (from *S. cerevisiae*, M7504), all purchased from Sigma-Aldrich (St. Louis, MO, USA).

### 4.2. RNA Isolation and RNA-seq

*E. coli*, *S. aureus*, and *C. albicans* were resuspended in insect saline and adjusted to a final concentration of 1 × 10^8^ CFU/mL. The suspensions of these three microorganisms were mixed in equal volumes and injected into 2nd-instar *A. pernyi* larvae. Larvae injected with an equal volume of insect saline served as the control group. Each group consisted of six larvae, with three biological replicates per condition. After incubation for 9 h, total RNA was extracted using TRIzol reagent (Thermo Scientific™, Waltham, MA, USA). Transcriptome sequencing of *A. pernyi* samples was performed using the Illumina (Illumina Inc., San Diego, CA, USA) sequencing platform and the paired-end sequencing method at Beijing, China Nuohe Zhiyuan Technology Co., Ltd. Differentially expressed genes (DEGs) were identified based on a fold change (FC) ≥ 4 and a *p*-value ≤ 0.001. Gene Ontology (GO) functional enrichment analysis was conducted, and molecular function-related factors were analyzed using GraphPad Prism 6 software. RNA-seq data visualization was performed using the Integrative Genomics Viewer (IGV, version 2.12.3) genome browser. GO functional enrichment analysis of DEGs was conducted using the OmicStudio tool (https://www.omicstudio.cn/tool, accessed on 28 April 2025).

### 4.3. Sequence Analysis and Alignment

Amino acid sequence alignments of *Ap*-proHP6 (GenBank ID: ANJ42865.1) and immune-related serine proteases from multiple species were performed using the MAFFT 7.0 program (https://mafft.cbrc.jp/) and MEGA 11 software. Phylogenetic trees were constructed in MEGA 11 using the neighbor-joining method with 1000 bootstrap replicates based on Poisson correlation models.

The three-dimensional (3D) structure of *Ap*-proHP6 was predicted using AlphaFold3 (https://golgi.sandbox.google.com/). Structural comparisons were performed using UCSF Chimera 1.17 software.

### 4.4. Isolation and Purification of Recombinant Protein and Preparation of the Polyclonal Antibody

The mature *Ap*-proHP6 gene fragment was digested with *Nco*I and *Xho*I (TaKaRa, Tokyo, Japan) and inserted into the pET-28a (+) expression vector. The recombinant expression vector pET-28a (+)-*Ap*-proHP6-His_6_ was constructed and transformed into *E. coli* (strain *DE3*). The induction protocol followed the methodology of He et al. [47]. r*Ap*-proHP6-His_6_ was purified using a Ni-Sepharose column (GE Healthcare, Beijing, China) with an imidazole gradient in the range of 50–500 mM. After dialysis, polyclonal antibodies were generated in rabbits using purified r*Ap*-proHP6-His_6_ following the protocol of He et al. [47]. Western blot analysis was conducted to verify the presence of both recombinant and native *Ap*-proHP6 proteins. For immunoblotting, we followed the protocol of He et al. [47]. Differently, after blocking the polyvinylidene difluoride (PVDF) membranes (Millipore, Bedford, MA, USA), it was incubated with rabbit anti-*Ap*-proHP6-His_6_ polyclonal antibody (1:4000 dilution) for 1.5 h. The primers used in this study are listed in Appendix A. All primers were synthesized by BGI Genomics Co., Ltd. (Shenzhen, China).

r*Ap*-proHP6-Tb-His_6_ was prepared using the same methodology. The *Ap*-proHP6 cleavage site was predicted using the Conserved Domain Database (https://www.ncbi.nlm.nih.gov/Structure/cdd/wrpsb.cgi (accessed on 8 February 2025)), and primers were designed to introduce a thrombin (Tb) cleavage site (Appendix A). The mutant sequence *Ap*-proHP6-Tb was obtained, and r*Ap*-proHP6-Tb-His_6_ was purified using the same protocol as r*Ap*-proHP6-His_6_.

### 4.5. Expression Properties in Response to Pathogen Injection

*E. coli*, *S. aureus*, and *C. albicans* were diluted to 1 × 10^8^ CFU/mL in insect saline and injected into fifth-instar *A. pernyi* larvae. Larvae injected with an equal volume of insect saline served as negative controls, with six larvae per group. After 6 h, larvae were anesthetized by freezing (0~4 °C) and dissected to collect the fat body, midgut, epidermis, Malpighian tubules, and hemocytes. Total RNA was extracted from each tissue sample using TRIzol reagent following the protocol of Duan et al. [50]. Reverse transcription was performed to obtain cDNA, and RT-qPCR (the SYBR Premix Ex Taq™ real-time PCR kit, TaKaRa, Tokyo, Japan; CFX96 Touch, Bio-Rad Laboratories, Inc., Hercules, CA, USA) was conducted. Relative gene expression was determined using the 2^−ΔΔCt^ method. Similarly, the temporal expression profile of *Ap-proHP6* after whole-larva injection was examined by RT-qPCR according to the protocol of Duan et al. [50,51], except that sampling times were 0, 3, 6, 9, 12, 18, 24, 36, 48, and 72 h post-injection. At each time point, hemolymph was collected, and *Ap*-proHP6 protein levels were analyzed via Western blot as described in Section 4.4. The primers used in this assay are listed in Appendix A.

### 4.6. Detection of RNAi Efficiency

Double-stranded RNA (dsRNA) targeting *Ap-proHP6* or *enhanced green fluorescent protein (EGFP)* was synthesized with the T7 promoter sequence. The synthesized dsRNA was separately injected into the second-instar larvae. In addition to injecting 200 ng (5 μL, 40 ng/μL) of dsRNA into each second-instar larva, while larvae injected with insect saline served as negative controls, following He et al.’s experimental protocol [47]. Gene expression changes were assessed by RT-qPCR at multiple time points (0–72 h post-injection), and corresponding protein level changes were analyzed via immunoblotting, as detailed in Section 4.4 and 4.5.

### 4.7. Transcription Level of AMPs in Response to Ap-HP6 Depletion

To assess the impact of *Ap*-HP6 depletion on AMP expression following bacterial invasion, 5 μL of each microbial suspension (2 × 10^8^ CFU/mL in insect saline) was injected into RNAi-treated larvae. Larvae injected with insect saline instead of microbial cells served as controls. After 9 h, total RNA was extracted from whole larvae, followed by cDNA synthesis. RT-qPCR was performed to detect the relative expression of selected antimicrobial peptides, as described in Section 4.5, using AMP-specific primers listed in Appendix A.

### 4.8. PO Activity Assay

The role of endogenous *Ap*-HP6 in the PPO activation cascade was assessed following a previously described method with minor modifications. Briefly, *Ap*-proHP6-deficient hemolymph after RNAi was collected. The method of He et al. was followed to determine PO [47]. Furthermore, the effect of endogenous HP6 protein on PPO activation was investigated. To generate HP6-deficient plasma, 10 mL of purified anti-r*Ap*-proHP6-His_6_ antibody (0.8 mg/mL) was pre-incubated with 90 mL of native hemolymph at 25 °C for 30 min. Hemolymph pre-incubated with an equal volume of buffer A served as the negative control. PO activity in response to various irritants was then evaluated as described above.

To examine the effect of exogenous r*Ap*-HP6 on PPO activation, r*Ap*-proHP6-Tb-His_6_ (150 μg/mL) was dialyzed against buffer B (20 mM Tris-HCl, 100 mM NaCl, 2 mM CaCl_2_, pH 8.0) and incubated with thrombin at a molar ratio of 1:70 at 23 °C for 6 h. The hydrolyzed r*Ap*-proHP6-Tb-His_6_ was then dialyzed against buffer A and designated as r*Ap*-proHP6-Tb-His_6_+Tb. A mixture of 10 μL of natural hemolymph and 10 μL of stimulant was separately pre-incubated with 10 μL of buffer A, 10 μL of thrombin, 10 μL of r*Ap*-proHP6-His_6_, 10 μL of *Ap*-proHP6-Tb-His_6_, or 10 μL of *Ap*-proHP6-Tb-His_6_+Tb at 25 °C for 5 min. Samples without stimulants (using buffer A as a substitute) served as controls. PO activity in response to various irritants was then evaluated.

### 4.9. Determined Enzymatic Catalysis Conditions

#### 4.9.1. Determination of the Catalytic Activity of r*Ap*-HP6

To assess whether r*Ap*-HP6, produced via the *thrombin*-mediated hydrolysis of r*Ap*-proHP6-Tb-His_6_, retains serine protease catalytic activity, the hydrolyzed r*Ap*-HP6 was dialyzed against buffer A. A reaction mixture containing 20 μL of r*Ap*-HP6, 30 μL of substrate S-2288 (a chromogenic substrate for broad-spectrum serine proteases, AS00-0104, Beijing, China Asnail Biotechnology Co., Ltd., brand: Asnail), and 150 μL of buffer A was prepared to a total volume of 200 μL [52]. Negative controls included samples lacking r*Ap*-proHP6-Tb-His_6_, thrombin, or both (substituted with buffer A), as well as samples where r*Ap*-proHP6-His_6_ was used in place of r*Ap*-proHP6-Tb-His_6_. After incubation at 37 °C for 30 min, absorbance at 405 (OD_405_) nm was measured using a microplate reader (Thermo Scientific). One unit of enzymatic activity was defined as an increase of 0.01 in absorbance at OD_405_ per min.

#### 4.9.2. Effect of Time on the Enzyme Activity of r*Ap*-HP6

Following the procedure described in Section 4.9.1, absorbance at OD_405_ was continuously monitored over a 60 min period to evaluate the time-dependent enzymatic activity of r*Ap*-HP6.

#### 4.9.3. Effect of pH on the Enzyme Activity of r*Ap*-HP6

The hydrolyzed r*Ap*-HP6 was dialyzed against 13 pH gradients ranging from 4.0 to 10.0 (20 mM Tris-HCl, with 0.5-unit pH intervals). After 8 h of dialysis, samples were incubated at 37 °C for 15 min, and absorbance at OD_405_ was measured as described in Section 4.9.1.

#### 4.9.4. Effect of Temperature on the Enzyme Activity of r*Ap*-HP6

The hydrolyzed r*Ap*-HP6 was dialyzed against buffer C (20 mM Tris-HCl, pH 8.0) and incubated at various temperatures ranging from 20 °C to 70 °C, with 5 °C intervals, for 15 min following the procedure in Section 4.9.1. Absorbance at OD_405_ was then recorded for each condition.

### 4.10. Statistical Analysis

All experiments were performed at least three times, yielding consistent results. Each bar in the figures represents the mean ± standard deviation (SD) ( *N*= 3). The effects of *Ap*-HP6 on AMP synthesis and PO activity were analyzed using analysis of variance (One-Way ANOVA), followed by Tukey’s multiple comparisons test.

## 5. Conclusions

In conclusion, these findings demonstrate that *Ap*-HP6 serves as an essential component of the immune-associated serine protease cascade in *A. pernyi* hemolymph. It plays a pivotal role in pathogen-mediated PPO activation and contributes to the induction of AMP synthesis. Furthermore, in vitro hydrolysis of r*Ap*-proHP6-Tb-His_6_ yielded active r*Ap*-HP6 exhibiting serine protease activity, with optimal reaction conditions for substrate S-2288 hydrolysis identified as pH 8.0 at 50 °C for 15 min.

## Figures and Tables

**Figure 1 ijms-26-04514-f001:**
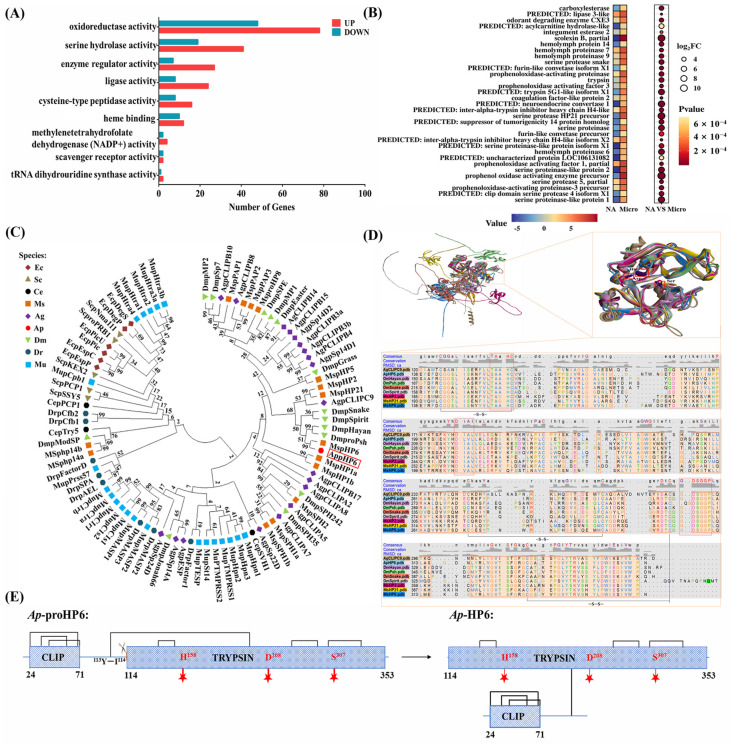
Bioinformatics analysis of *Ap*-proHP6. (**A**) GO annotation of differentially expressed genes (DEGs) in molecular functional categories after 12 h of microbial infection of the second-instar larvae of *A. pernyi*. Up and down represent the upregulated and downregulated DEGs, respectively. (**B**) Heat map and bubble map of the expression differences of 33 significantly upregulated serine hydrolases after microbial induction. (**C**) Phylogenetic tree of serine proteases of various organisms. Phylogenetic tree constructed on the basis of 89 amino acid sequences (Appendix A—amino acid sequences of serine proteases). Ec: *E. coli*, Sc: *S. cerevisiae*, Ce: *C. elegans*, Ms: *M. sexta*, Ag: *A. gambiae*, Ap: *A. pernyi*, Dm: *D. melanogaster*, Dr: *Danio rerio*, and Mu: mouse. Ap-proHP6 is marked with a red box. (**D**) Structure-based sequence alignment and the highly conserved spatial structures were visualized with UCSF chimera 1.17. The tertiary structure model of each protein is represented by one color respectively and corresponds to the color of each protein name below. Based on the spatial structure sequence alignment, it is used to distinguish the spatial structure of each protein. Amino acid sequences contained in each of the 9 proteins are highlighted as light-orange regions. And the heading “RMSD: ca” shows the spatial variation in each column (root mean square deviation of α-carbon) in the form of a histogram. Conservation and consensus are indicated above the alignment and conserved disulfide bridges are represented as a black line. Conserved sequences surrounding the catalytic triad are labeled with a red box. (**E**) Predicted domain changes in *Ap*-proHP6 from zymogen hydrolysis to enzyme *Ap*-HP6. The red five-pointed stars represent the catalytic active sites of *Ap*-HP6. *Ap*-proHP6 belongs to the CLIP subfamily, with an N-terminal clip domain, followed by a linker region and a C-terminal serine protease domain characteristic of the trypsin family. A disulfide bond (^99^S-S^228^) connects the linker region to the trypsin domain. Upon specific proteolytic cleavage at position ^113^Y-I^114^, *Ap*-proHP6 is activated, and despite cleavage, the clip and trypsin domains remain covalently linked to form the active enzyme *Ap*-HP6.

**Figure 2 ijms-26-04514-f002:**
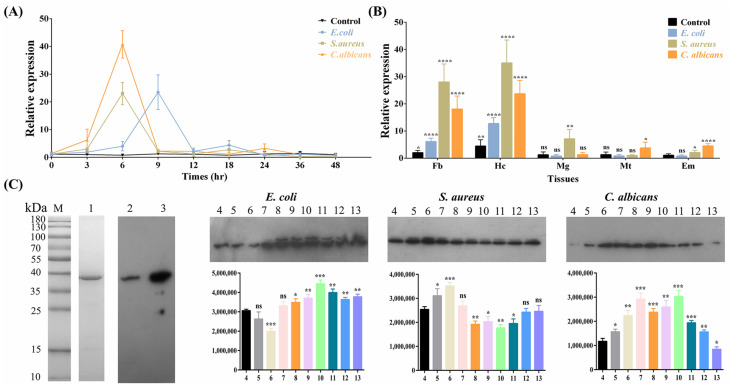
Expression profile of *Ap*-HP6 in the immune response of *A. pernyi*. (**A**) Time–course analysis of *Ap*-proHP6 mRNA expression in the second-instar larva after immune challenge with *S. aureus*, *E. coli*, and *C. albicans* (five individuals per time point). (**B**) Tissue distribution of *Ap*-proHP6 mRNA in response to stimulation with microbes at the same immune challenge time 6 h was also detected. Tissues involved in the experiment were as follows: Em, epidermis; Fb, fat body; Mg, midgut; Mt, Malpighian tube; and Hc, hemocyte. Each tissue was collected before injection (control) or 6 h post-injection with microbes (induced). (**C**) SDS-PAGE and immunoblot analysis of *Ap*-proHP6. Lane 1: purified r*Ap*-proHP6-His_6_ (7.4 μg, Coomassie Brilliant Blue staining); Lane 2: *Ap*-proHP6 in natural *A. pernyi* hemolymph (13 mg/mL); Lane 3: purified r*Ap*-proHP6-His_6_ (20 μg/mL); Lane 4: natural hemolymph; 5: Hemolymph collected 3 h after microbial injection; 6: Hemolymph collected 6 h after microbial injection; 7: Hemolymph collected 9 h after microbial injection; 8: Hemolymph collected 12 h after microbial injection; 9: Hemolymph collected 18 h after microbial injection; 10: Hemolymph collected 24 h after microbial injection; 11: Hemolymph collected 36 h after microbial injection; 12: Hemolymph collected 48 h after microbial injection; 13: Hemolymph collected 72 h after microbial injection. Lanes 4–13: Hemolymph samples at a concentration of 10 mg/mL. Lanes 2–13 (15 μL volume per lane) were detected by Western blot using anti-r*Ap*-proHP6-His_6_ antibody; the results represent 3 biological replicates and the gray-scale statistical is shown below. Each bar represents the mean ± SD (*N* = 3). mRNA, messenger RNA; SD, standard deviation. *: *p* < 0.05, **: *p* < 0.01, ***: *p* < 0.001, and ****: *p* < 0.0001; ns: no significant difference (Student’s *t*-test).

**Figure 3 ijms-26-04514-f003:**
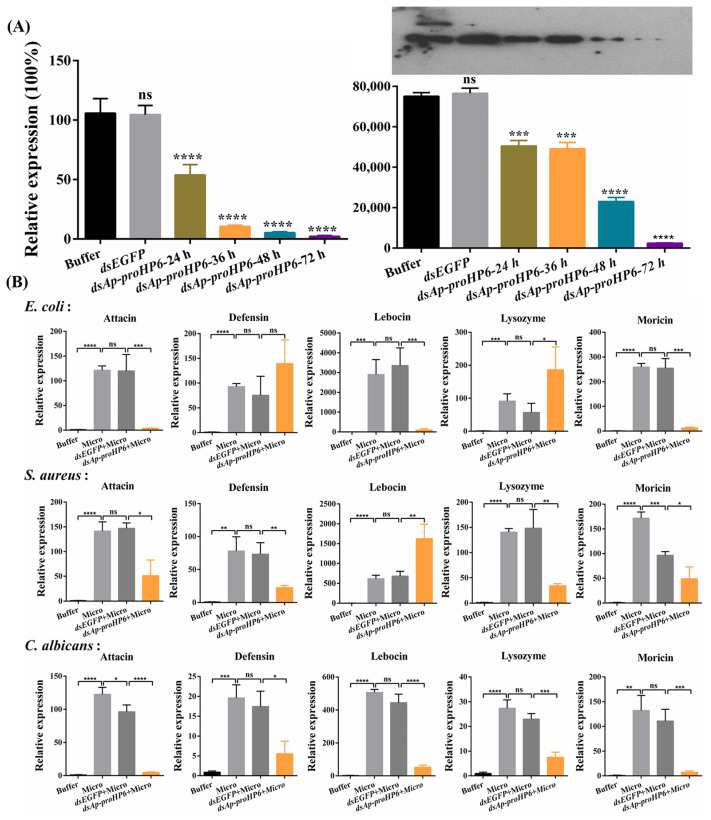
Effect of *Ap*-HP6 on the induction of AMP synthesis in *A. pernyi* larvae by RT-qPCR analysis. (**A**) RNAi efficiency estimation by RT-qPCR and Western blot. For RT-qPCR, whole-bodied larvae were collected post-injection with ds*Ap-proHP6* (24 h, 36 h, 48 h, and 72 h) or ds*EGFP* (48 h) double-stranded RNA. For Western blot analysis, the level of *Ap*-proHP6 protein decline in *A. pernyi* hemolymph was evaluated using an anti-r*Ap*-proHP6-His_6_ polyclonal antibody and each lane was assessed by the gray-level analysis. A total of 10 mg of plasma samples from ds*Ap*-proHP6-treated and control larvae was used. Buffer: larvae injected with buffer (distilled water after DEPC treatment); ds*EGFP* and ds*Ap-proHP6*: larvae injected with dsRNA of EGFP and *Ap*-proHP6, respectively. The results represent 3 biological replicates and the gray-scale statistical result is shown below. (**B**) Relative mRNA expression levels of antibacterial peptides in response to microbial stimulation. Micro: larvae collected 9 h after injection with formaldehyde-killed microbial cells (10^5^ cells per larva); ds*EGFP*+Micro: larvae were first injected with ds*EGFP* for 48 h and then immune-challenged with microbial cells for another 9 h; ds*Ap-proHP6*+Micro: larvae were first injected with ds*Ap-proHP6* for 48 h and then immune-challenged with microbes for another 9 h. Microbes: *E. coli*, *S. aureus*, and *C. albicans*. The relative expression level of antibacterial peptides detected by RT-qPCR: attacin, defensin, lebocin, lysozyme, and moricin. Each bar represents the mean ± SD (*N* = 3). *: *p* < 0.05, **: *p* < 0.01, ***: *p* < 0.001, ****: *p* < 0.0001, and ns: no significant difference.

**Figure 4 ijms-26-04514-f004:**
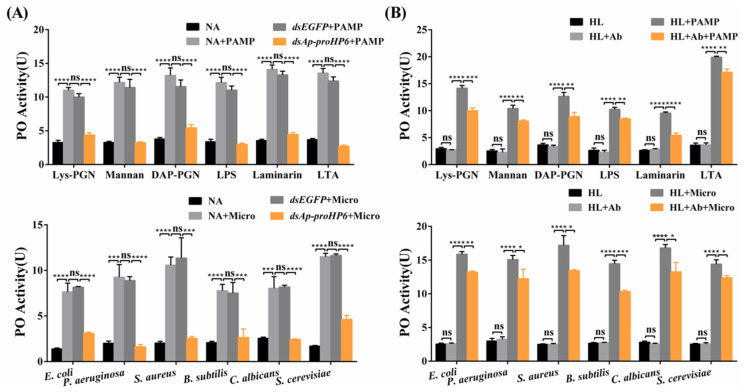
Contribution of *Ap*-HP6 to PPO activation. (**A**) The effects of endogenous *Ap*-HP6 on PPO cascade activation by irritants (microbes/PAMPs) were investigated. Hemocyte-free plasma of 2nd-instar larvae was collected 48 h after dsRNA injection. Then, the PO activity triggered by irritants was examined. NA: larvae without treatment; ds*EGFP* and ds*Ap-proHP6*: larvae injected with dsRNA of EGFP and *Ap*-proHP6, respectively. (**B**) Irritants dependent PPO activation in *Ap*-proHP6 blocked plasma. Hemocyte-free plasma samples were pre-incubated with anti-r*Ap*-proHP6-His_6_ antibody and Buffer A. PO activity of these samples triggered by microbes and soluble PAMPs was examined and compared. HL: native cell-free hemolymph (total protein: 12 mg/mL). PAMP: laminarin, Lys-PGN, DAP-PGN, LTA, LPS, and mannan (125 ng per sample). Ab: anti-r*Ap*-proHP6-His_6_ (0.8 mg/mL). Micro: microorganism, *E. coli*, *S. aureus*, *C. albicans*, *B. subtilis*, *P. aeruginosa*, and *S. cerevisiae*. Each bar represents the mean ± SD (*N* = 3); *: *p* < 0.05, **: *p* < 0.01, ***: *p* < 0.001, and ****: *p* < 0.0001; ns: no significant difference.

**Figure 5 ijms-26-04514-f005:**
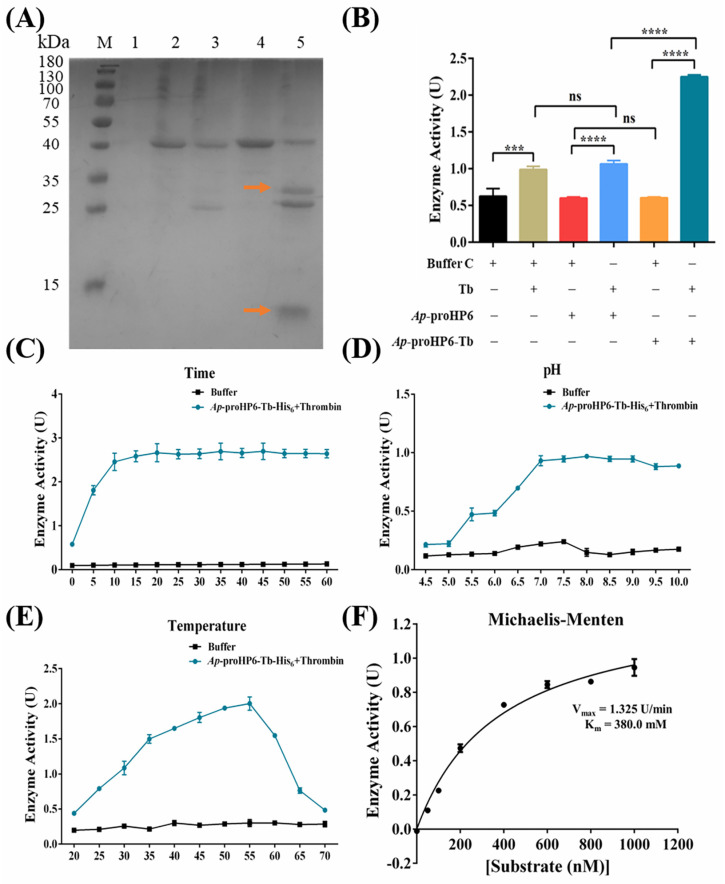
Serine protease enzymatic activity assay of r*Ap*-HP6. (**A**) SDS-PAGE analysis was performed to validate the thrombin-mediated cleavage of the r*Ap*-proHP6-Tb-His_6_. Lane 1: thrombin (1 μg); Lane 2: r*Ap*-proHP6-His_6_ (4 μg); Lane 3: r*Ap*-proHP6-His_6_+Tb; Lane 4: r*Ap*-proHP6-Tb-His_6_ (4 μg); Lane 5: r*Ap*-proHP6-Tb-His_6_+Tb. The orange arrows represent the two protein bands produced by thrombin cleavage at the thrombin cleavage site of r*Ap*-proHP6-Tb-His_6_. (**B**) The serine protease catalytic activity of r*Ap*-HP6 was determined based on the substrate S-2288. (**C**–**E**) Effects of pH, temperature, and time on the enzymatic reaction of r*Ap*-HP6 to substrate S-2288. *N* = 3 independent experiments. (**F**) Michaelis–Menten equation based on substrate S-2288 and absorbance. The horizontal axis represents different concentrations of substrate S-2288 ranging from 0 nM to 1.0 mM (0 nM, 50 nM, 100 nM, 200 nM, 400 nM, 600 nM, 800 nM, and 1000 nM). The vertical axis depicts the change in absorbance of *Ap*-HP6 at OD_405_ after the hydrolysis of the substrate. The kinetic parameters, V_max_ and K_m_, were estimated by fitting various parameters using the Michaelis–Menten equation. The specific parameter values used for this plot are V_max_ = 1.325 U/min and K_m_ = 380.0 nM. Each bar represents the mean ± SD (*N* = 3); ***: *p* < 0.001, and ****: *p* < 0.0001; ns: no significant difference.

**Figure 6 ijms-26-04514-f006:**
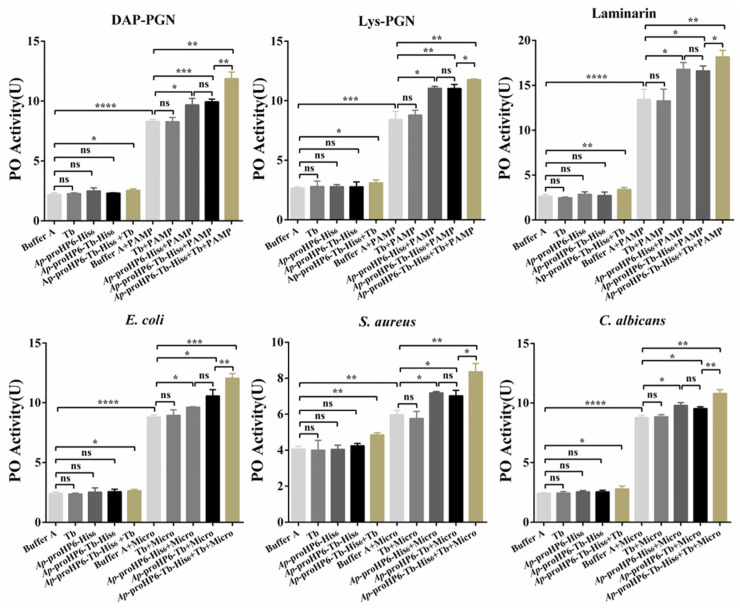
The effect of exogenous r*Ap*-HP6 on PPO activity triggered by irritants. HL, native hemolymph (total protein: 12 mg/mL); r*Ap*-proHP6-His_6_ (20 μg/mL), r*Ap*-proHP6-Tb-His_6_ (20 μg/mL); PAMP: laminarin, Lys-PGN, and DAP-PGN (125 ng per sample), Micro: microorganism, *E. coli*, *S. aureus*, and *C. albicans*. Each bar represents the mean ± SD (*N* = 3); *: *p* < 0.05, **: *p* < 0.01, ***: *p* < 0.001, and ****: *p* < 0.0001; ns: no significant difference (Student’s *t*-test).

## Data Availability

The data presented in this study are available from the corresponding authors upon reasonable request.

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
