# Peer review of "Unveiling the Multifaceted Role of HP6: A Critical Regulator of Humoral Immunity in Antheraea pernyi (Lepidoptera: Saturniidae)"

_ijms, 2025, doi:10.3390/ijms26104514_

Round 1
Reviewer 1 Report
Comments and Suggestions for Authors
The manuscript looks interesting, but some modifications are recommended.
- The title is somewhat prosaic. I suggest modifying the title to be more expressive and convenient.
- The introduction is too long, I suggest shortening it.
- In Sequence analysis and alignment, what is your reference for performing 1,000 bootstrap replicates?
- Remove Table (1) to supplementary materials.
- Clarify how did you designated the primers' sequence, and also clarify from where you obtained them (source company).
- In Sequence analysis and alignment, why the insect saline was specifically injected into the fifth-instar of A. pernyi larvae? and later in lines 640 and 641 you injected the second-instar larva?!
- In line 627, anesthetized by freezing, please clarify freezing to which temperature.
- In RT-qPCR, did the authors used Syber green or Taqman for the amplification step? clarify this point in the manuscript.
- Which gene was used as a reference gene for western blot and also for RT-qPCR?
- From where you obtained substrate S-2288?
- What is your reference for Determination of the Catalytic Activity of rAp-HP6, or you used a commercially available kits?
- Which type of ANOVA was used? one-way or two-way? clarify.
- The flow of the methodology is confusing to some extent. I Kindly make a flowchart for your work to be clearer and more organized for readers.
- Did the author obtained any ethical approval for this study?
- The resolution of Fig. 1 is low and the writing font can't be seen. please provide clearer figure with larger font.
- The authors have to provide a conclusion at the end of the manuscript as a separate heading.
Author Response
Comments 1: The title is somewhat prosaic. I suggest modifying the title to be more expressive and convenient.
Response 1: Agree. The title has been rephrased to better reflect the study. “Unveiling the Multifaceted Role of HP6: A Critical Regulator of Humoral Immunity in Antheraea pernyi (Lepidoptera: Saturniidae)” (Page1, Lines 2 ~ 4).
Comments 2: The introduction is too long, I suggest shortening it.
Response 2: Agree. The Introduction section has been condensed to improve focus, and the manuscript text has been revised for clarity. The updated version has been resubmitted.
Comments 3: In Sequence analysis and alignment, what is your reference for performing 1,000 bootstrap replicates?
Response 3: We acknowledge the significance of this question raised by the reviewer. The selection of performing 1,000 bootstrap replicates in sequence analysis and alignment is grounded in statistical principles, field-specific conventions, and support from seminal literature.
(1) Statistical Rationale
The bootstrap method estimates the distribution of statistical metrics through resampling. Increasing the number of replicates enhances result stability, with 1,000 replicates striking a balance between computational feasibility and statistical accuracy. This ensures sufficiently small standard errors for support values (typically <1 %).
(2) Field Consensus
Leading journals in phylogenetics and molecular evolution (e.g., Molecular Biology and Evolution) commonly recommend or mandate 1,000 bootstrap replicates to ensure result reliability, reflecting established best practices in the discipline.
(3) Support from Seminal Literature.
Hillis, D.M.; Bull, J.J. An Empirical Test of Bootstrapping as a Method for Assessing Confidence in Phylogenetic Analysis. Systematic Biology 1993, 182-192, doi: 10.1093/sysbio/42.2.182. This reference demonstrates that 1,000 bootstrap replicates are generally sufficient for robust confidence interval estimation.
Moreover, this approach has been widely adopted across disciplines, as exemplified by the following applications:
â‘ Zhang, H.; Gao, J.; Ma, Z.; Liu, Y.; Wang, G.; Liu, Q.; Du, Y.; Xing, D.; Li, C.; Zhao, T.; et al. Wolbachia infection in field-collected Aedes aegypti in Yunnan Province, southwestern China. Frontiers in cellular and infection microbiology 2022, 12, 1082809, doi:10.3389/fcimb.2022.1082809.
â‘¡ Liu, Q.; Shan, Q. Associations of α-linolenic acid dietary intake with very short sleep duration in adults. Frontiers in public health 2022, 10, 986424, doi:10.3389/fpubh.2022.986424.
Comments 4: Remove Table (1) to supplementary materials.
Response 4: Agree. Table 1 has been moved to the Supplementary Material and is now designated as Table S1. And it has been resubmitted. The relevant information has been revised accordingly in Section 4.4 (Page 17, Line 600; Page 17, Line 605), Section 4.5 (Page 17, Line 621), and Section 4.7 (Page 17, Line 638).
Comments 5: Clarify how did you designated the primers' sequence, and also clarify from where you obtained them (source company).
Response 5: We appreciate your thoughtful suggestion.
(1) Designated the Primers' sequence
â‘ Construction of recombinant plasmid pET-28a(+)-Ap-proHP6-His6. The signal peptide of Ap-proHP6 was predicted and removed. Forward and reverse primers were designed based on the full-length coding sequence of the mature Ap-proHP6.
â‘¡ Construction of recombinant plasmid pET-28a(+)-Ap-proHP6-Tb-His6. The cleavage site of Ap-proHP6 was predicted using the Conserved Domain Database (https://www.ncbi.nlm.nih.gov/Structure/cdd/wrpsb.cgi). Primers were designed to incorporate a thrombin cleavage site at this position via site-directed mutagenesis. Simultaneously, primers targeting the full-length mature Ap-proHP6 sequence were designed.
â‘¢ Primers were designed for dsRNA synthesis targeting the gene of interest. Genomic regions of approximately 600 bp, devoid of intronic sequences, were selected and subjected to BLASTn analysis (NCBI) to eliminate homologous sequences with other genes, thereby minimizing potential off-target effects. Primer pairs were subsequently designed for regions fulfilling both length requirements and sequence specificity criteria.
â‘£ Primers were designed for RT-qPCR analysis by selecting ~150 bp regions within target genes. Candidate sequences underwent rigorous specificity validation through BLAST alignment (NCBI) to eliminate cross-reactivity with homologous genomic regions. Primer pairs were subsequently synthesized only for genomic loci satisfying both amplicon length constraints and sequence uniqueness criteria.
(2) Primer Synthesis
All primers were synthesized by BGI Genomics Co., Ltd. (Shenzhen, China). The pertinent information has been suitably modified within Section 4.4 (Page 17, Lines 600 and 601).
Comments 6: In Sequence analysis and alignment, why the insect saline was specifically injected into the fifth-instar of A. pernyi larvae? and later in lines 640 and 641 you injected the second-instar larva?!
Response 6: We gratefully recognize the reviewer's insightful comment on this issue.
First, we sincerely apologize for the inadvertent errors identified in our manuscript. And we revised the title of Section 4.5 (Page 17, Line 698) to "4.5 Expression Properties in Response to Pathogen Injection".
(1) In lines 609 and 610: Injection of fifth-instar larvae with insect saline (control group) and analysis of Ap-proHP6 mRNA expression across tissues.
Fifth-instar larvae were selected due to their larger body size, which facilitates efficient tissue dissection. Additionally, tissues at this developmental stage exhibit higher functional maturity: Hemocyte populations are more abundant and functionally developed; Fat body undergoes significant expansion (accounting for >30 % of body weight) and reaches peak metabolic activity.
(2) In Lines 626 and 627 (The original manuscript is in lines 640 and 641), regarding the experimental design of using RNAi technology to examine the effect of HP6 on the synthesis of antimicrobial peptides (AMPs) in the whole body of the insect, our choice to inject insect saline into second-instar larvae (control group) was based on the following considerations:
â‘ Feasibility of experimental design: We aimed to analyze systemic changes in AMPs across the entire body of A. pernyi larvae. In order to reduce individual differences, we will use a group of multiple larvae together with liquid nitrogen treatment and grinding. The second instar larvae are small and easy to operate. However, the fifth-instar larvae are very large, and it is difficult to treat and grind multiple larvae together with liquid nitrogen.
â‘¡ Uniformity of microbial injection dosage: Second-instar larvae, with their smaller size and better size consistency, allow precise control over the injected volume of microbial suspension, thereby minimizing inter-individual variability. In contrast, fifth-instar larvae have a higher body fluid content (hemolymph accounts for over 15 % of body weight) but exhibit significant size variation. This results in microbial dilution and inconsistent infection intensity across individuals.
Comments 7: In line 627, anesthetized by freezing, please clarify freezing to which temperature.
Response 7: We thank the reviewer for identifying this issue. A. pernyi is an ectothermic organism, and low temperatures inhibit its neural conduction and muscular activity. Exposure to 0 ~ 4 ℃ induces a reversible anesthetic state in larvae. Excessively low temperatures cause tissue damage from ice crystal formation, while higher temperatures fail to achieve complete anesthesia. The relevant content has been revised accordingly in Section 4.5 (Page 17, Line 612).
Comments 8: In RT-qPCR, did the authors used Syber green or Taqman for the amplification step? clarify this point in the manuscript.
Response 8: We appreciate your thoughtful suggestion. In RT-qPCR experiments, we utilized the SYBR Premix Ex Taq™ real-time PCR kit, with amplification steps performed using the SYBR Green detection method. The pertinent content has been suitably modified within Section 4.5 (Page 17, Line 615).
Comments 9: Which gene was used as a reference gene for western blot and also for RT-qPCR?
Response 9: We gratefully recognize the reviewer's insightful comment on this issue.
(1) In RT-qPCR experiments, 18S rRNA was used as the internal reference gene to normalize the expression levels of target genes.
(2) In western blot experiments, no internal reference protein was employed, for the following reasons:
â‘ Sample Specificity: A. pernyi hemolymph contains high-abundance lipoproteins. Preliminary experiments revealed insufficient expression stability of conventional internal references (β-actin/GAPDH) (coefficient of variation, CV > 30%), which may be linked to immune stimulation or sample processing. As reported in the reference (Ve, H.; Cabana, V.C.; Gouspillou, G.; Lussier, M.P. Quantitative Immunoblotting Analyses Reveal that the Abundance of Actin, Tubulin, Synaptophysin and EEA1 Proteins is Altered in the Brains of Aged Mice. Neuroscience 2020, 442, 100-113, doi:10.1016/j.neuroscience.2020.06.044.), studies have shown significant fluctuations in β-actin and β-tubulin expression across different brain tissues during aging in Mus musculus, indicating that traditional internal references may lack stability under dynamic physiological conditions, particularly in highly heterogeneous samples (e.g., body fluids with high-abundance interfering proteins). These findings align with the challenges observed in A. pernyi hemolymph, supporting the unsuitability of conventional internal references for complex samples.
â‘¡ Application of Total Protein Normalization: During the experimental process, the total protein concentration in A. pernyi hemolymph was quantified using the BCA assay and normalized to 8 mg/mL, with a strictly controlled loading volume of 20 μL. Furthermore, western blot results demonstrated consistent trends in HP6 expression with RT-qPCR data, validating the reliability of this approach.
â‘¢ Literature support:
Duan, X.; Fu, T.; Liu, C.; Wang, F.; Liu, C.; Zhao, L.; Yu, J.; Wang, X.; Zhang, R. The role of a novel secretory peptidoglycan recognition protein with antibacterial ability from the Chinese Oak Silkworm Antheraea pernyi in humoral immunity. Insect biochemistry and molecular biology 2024, 171, 104151, doi:10.1016/j.ibmb.2024.104151.
He, X.; Zhou, T.; Cai, Y.; Liu, Y.; Zhao, S.; Zhang, J.; Wang, X.; Zhang, R. A Versatile Hemolin With Pattern Recognitional Contributions to the Humoral Immune Responses of the Chinese Oak Silkworm Antheraea pernyi. Frontiers in immunology 2022, 13, 904862, doi:10.3389/fimmu.2022.904862.
In the above two research papers on the humoral immune response of A. pernyi, the western blot experiment also adopted a method without internal reference protein.
Comments 10: From where you obtained substrate S-2288?
Response 10: We thank the reviewer for identifying this issue. S-2288 was purchased from Beijing Asnail Biotechnology Co., Ltd., Code: AS00-0104, brand: Asnail. In response to reviewer feedback, the relevant content has been amended in Section 4.9.1 (Page 18, Line 663).
Comments 11: What is your reference for Determination of the Catalytic Activity of rAp-HP6, or you used a commercially available kits?
Response 11: We express our gratitude to the reviewer for bringing this issue to our attention.
No commercial kits were used during the experiments. The reference materials are as follows:
(1) Rationale for Substrate Selection: S-2288 (H-D-Ile-Pro-Arg-pNA) is a chromogenic substrate for broad-spectrum serine proteases. Its design is based on the cleavage specificity of serine proteases for defined peptide bonds. The substrate exhibits a characteristic absorption peak at 405 nm, enabling direct quantification of enzymatic activity via spectrophotometric analysis.
(2) Chromogenic Mechanism: Upon substrate cleavage, p-nitroaniline (pNA) is released. The free form of pNA exhibits a characteristic absorption peak at 405 nm, allowing direct spectrophotometric quantification of enzymatic activity.
(3) Literature support:
McCarthy, J.R.; Sazonova, I.Y.; Erdem, S.S.; Hara, T.; Thompson, B.D.; Patel, P.; Botnaru, I.; Lin, C.P.; Reed, G.L.; Weissleder, R.; et al. Multifunctional nanoagent for thrombus-targeted fibrinolytic therapy. Nanomedicine (London, England) 2012, 7, 1017-1028, doi:10.2217/nnm.11.179. And this work has been added to the reference list as entry [52] in Section 4.9.1 (Page 18, Line 664).
Comments 12: Which type of ANOVA was used? one-way or two-way? clarify.
Response 12: We thank the reviewer for pointing out this issue to us. The type of ANOVA is One-Way ANOVA: A single independent variable that compares differences between groups. We have made revisions to the content in Section 4.10 (Page 18, Lines 687 and 688).
Comments 13: The flow of the methodology is confusing to some extent. I Kindly make a flowchart for your work to be clearer and more organized for readers.
Response 13: Agree. To illustrate the key components of this study, we have constructed the following flowchart:
Please see the attachment.
Comments 14: Did the author obtained any ethical approval for this study?
Response 14: We are thankful to the reviewer for highlighting this issue. This study has received ethical approval. The Institutional Review Board Statement has been incorporated into the manuscript (Page 19, Lines 707–709), and the ARRIVE checklist has been submitted to the journal in accordance with reporting guidelines.
Comments 15: The resolution of Fig. 1 is low and the writing font can't be seen. please provide clearer figure with larger font.
Response 15: Agree. We are grateful to the reviewer for identifying this critical oversight. We have revised the font sizes of Figure 1 in Section 2.1 (Page 5, Line 164), made clearer picture and re-uploaded it.
Comments 16: The authors have to provide a conclusion at the end of the manuscript as a separate heading.
Response 16: Agree. We have added "2.7 Conclusion" in Section 2.7 (Page 12, Line 393).

Reviewer 2 Report
Comments and Suggestions for Authors
The manuscript describes the serine protease-hemolymph protease 6 (HP6) of Antheraea pernyi, the Chinese oak silkworm on a regulatory effect on the prophenoloxidase (PPO) and Toll pathway-mediated antimicrobial peptide (AMP) synthesis.
The original idea of this study was published by Qie et al. (2023) (Serpin-4 Negatively Regulates Prophenoloxidase Activation and Antimicrobial Peptide Synthesis in the Silkworm, Bombyx mori. Int. J. Mol. Sci. 25(1):313). This study simply uses a similarly related insect to the silkworm to confirm the results in this paper. Interestingly, authors do not cite this paper.
Also, An et al. (2009) (Functions of Manduca sexta Hemolymph Proteinases HP6 and HP8 in Two Innate Immune Pathways. J. Biol. Chem. 284(29):19716-26.) reported the relevant results compared to this study, yet it is not cited.
In order to publish this study, this reviewer thinks that authors need to show how their study is different than these papers and why these results in the manuscript are important.
Figures in the manuscript are generally too small and not readable (particularly Figure 1).
Author Response
Comments 1: In order to publish this study, this reviewer thinks that authors need to show how their study is different than these papers (the reviewer recommended references) and why these results in the manuscript are important.
Response 1: We sincerely appreciate the reviewer's valuable comments. We have carefully studied the two articles you provided, which are closely related to our research, and have now included them as references in our paper: reference 33 in Section 1 (Page 3, Line 91) and reference 51 in Section 4.5 (Page 17, Line 618). Although the experimental approaches employed in these two studies have been referenced in our work, it is still necessary to study the A. pernyi serine protease HP6.
(1) The insect immune signaling pathways exhibit remarkable specificity and diversity. Although the innate immune signaling cascades in insects have not been fully elucidated, existing studies in model organisms such as M. sexta, B. mori, and T. molitor have revealed significant diversity in these cascades. For instance, in T. molitor, peptidoglycan recognition protein-SA (PGRP-SA) recognizes pathogen-associated molecular patterns (PAMPs), thereby initiating the MSP-SAE-SPE cascade reaction, which ultimately triggers melanization responses and antimicrobial peptide (AMP) production[1]. In contrast, M. sexta employs a distinct immune cascade initiated by Peptidoglycan recognition protein (PGRP) and Gram-negative binding protein (GNBP), activating a series of hemolymph proteinases to induce prophenoloxidase (PPO) activation and AMP synthesis[2]. While B. mori and A. pernyi both belong to the order Lepidoptera, they are phylogenetically distant, classified under the families Bombycidae and Saturniidae, respectively. Taking Hemolin as an example, Bm-Hemolin primarily mediates specific humoral immunity, whereas A.p-Hemolin, adapted to wild environments, exhibits broader-spectrum resistance and enhanced synergy with cellular immunity[3,4]. These differences reflect the divergence in domestication history, ecological pressures, and immune strategies between the two silkworm species.
(2) As a characteristic economic insect in China, there is still a significant gap in the research of serine proteinase-mediated immune regulatory network of A. pernyi. Although the amino acid sequence of Ap-proHP6 is known, its immune functions have not yet been reported. While the Ap-proHP6 and Ms-proHP6 exhibit high conservation (75.25 % amino acid sequence identity), differences also exist between them. Notably, their predicted cleavage sites differ: 111GLY↓IING117 for Ap-proHP6 and 110DLH↓ILGG116 for Ms-proHP6. Structural differences dictate functional divergence, suggesting that the immune roles of Ap-HP6 and Ms-HP6 may also differ accordingly.
(3) As described in Reference 33, the specific band intensity of Ms-proHP6 showed no significant changes after immune challenge, and Ms-proHP6 transcripts did not increase in the fat body or hemocytes. In contrast, our study demonstrates that both the gene and protein expression levels of Ap-proHP6 significantly increased during the 0–72 hr post-immune stimulation, with Ap-proHP6 transcripts markedly upregulated in both the fat body and hemocytes. Second, in Reference 33, the authors injected larvae with Ms-proHP6 protein, induced immune responses using M. luteus, and measured the transcriptional levels of four AMPs. In our work, however, we employed RNA interference (RNAi) to silence Ap-proHP6 expression, challenged larvae with three distinct microbial pathogens, and analyzed the transcriptional levels of six AMPs. Additionally, we clarified the involvement of Ap-HP6 in Toll pathway-mediated AMP synthesis by detecting the Toll signaling pathway-related protein Spätzle. Finally, the authors of Reference 33 only investigated the effect of Ms-proHP6 on phenoloxidase (PO) activation by adding recombinant Ms-proHP6 and Ms-proHP6-Xa proteins to plasma in vitro, using M. luteus as the sole stimulant. In contrast, our study comprehensively explored the regulatory role of Ap-HP6 through multiple approaches: â‘ in vitro addition of recombinant Ap-proHP6 and rAp-HP6 proteins; â‘¡ RNAi-mediated Ap-proHP6 silencing in vivo; â‘¢ blocking endogenous Ap-proHP6 protein activity; and â‘£ analyzing the impact of Ap-proHP6 knockdown on pathogen-induced cuticular melanization in fifth-instar larvae. Moreover, we utilized six pathogenic microorganisms and six PAMPs in our experiments. All results consistently demonstrated that Ap-HP6 positively regulates the prophenoloxidase-activating system (PPO-AS).
In summary, the immune functional outcomes of Ap-HP6 and Ms-HP6 indeed differ. Comparatively, this study provides a more systematic elucidation of the positive regulatory role of Ap-HP6 in the early humoral immune response of A. pernyi—specifically in PPO-AS and AMP synthesis—triggered by diverse pathogenic microorganisms. Given the marked specificity and diversity inherent in insect immune signaling pathways, our comprehensive investigation of Ap-HP6 not only establishes a theoretical foundation for deciphering the unique serine protease cascade network in A. pernyi but also offers a critical breakthrough in understanding the molecular basis of disease resistance in lepidopteran insects. Furthermore, it bridges the knowledge gap in innate immune signaling pathway research for wild ecological insects.
References
- Jiang, R.; Kim, E.H.; Gong, J.H.; Kwon, H.M.; Kim, C.H.; Ryu, K.H.; Park, J.W.; Kurokawa, K.; Zhang, J.; Gubb, D.; et al. Three pairs of protease-serpin complexes cooperatively regulate the insect innate immune responses. The Journal of biological chemistry 2009, 284, 35652-35658, doi:10.1074/jbc.M109.071001.
- Wang, Y.; Yang, F.; Cao, X. Hemolymph protease-5 links the melanization and Toll immune pathways in the tobacco hornworm, Manduca sexta. Proceedings of the National Academy of Sciences of the United States of America 2020, 117, 23581-23587, doi:10.1073/pnas.2004761117.
- He, X.; Zhou, T.; Cai, Y.; Liu, Y.; Zhao, S.; Zhang, J.; Wang, X.; Zhang, R. A Versatile Hemolin With Pattern Recognitional Contributions to the Humoral Immune Responses of the Chinese Oak Silkworm Antheraea pernyi. Frontiers in immunology 2022, 13, 904862, doi:10.3389/fimmu.2022.904862.
- Liu, X.; Li, J.; Wang, Q.; Xia, H.; Chen, K. Functional analysis of hemolin gene from silkworm, Bombyx mori - immune and development. ISJ-Invertebrate Survival Journal 2017, 14, 330-339, doi:10.25431/1824-307X/isj.v14i1.330-339.
Comments 2: Figures in the manuscript are generally too small and not readable (particularly Figure 1).
Response 2: Agree. We are grateful to the reviewer for identifying this critical oversight, which have revised the font sizes of Figures 1 ~ 6, made clearer pictures and re-uploaded them. The specific locations are as follows: Figure 1 in Section 2.1 (Page 5, Line 164), Figure 2 in Section 2.2 (Page 6, Line 217), Figure 3 in Section 2.3 (Page 8, Line 258), Figure 4 in Section 2.4 (Page 9, Line 300), Figure 5 in Section 2.5 (Page 11, Line 354), Figure 6 in Section 2.6 (Page 12, Line 387).

Round 2
Reviewer 1 Report
Comments and Suggestions for Authors
Thank you for your effort to enhance the manuscript and covering most of the comments but still these 2 comments need responses:
- Please mention and add the flowchart figure in the main text of the manuscript.
- The conclusion is very short and not inclusive, and the authors must move it after the methodology section.
Author Response
Comments 1: Please mention and add the flowchart figure in the main text of the manuscript.
Response 1: We appreciate your thoughtful suggestion. The flowchart has been placed to the Supplementary Materials, designated as Figure S5, and appropriately referenced in the manuscript. The revised Supplementary Materials file has been resubmitted to the editorial system. The pertinent information has been suitably modified within Section 3 (Page 13, Lines 405 and 406).
Comments 2: The conclusion is very short and not inclusive, and the authors must move it after the methodology section.
Response 2: We are grateful to the reviewer for identifying this critical oversight. We have removed the content of Section 2.7 (Page 12, lines 393–396) from the manuscript. Additionally, conclusions (Section 5, Page 18, Lines 686–692) have been appended following the “Materials and Methods” section to enhance structural coherence:
“5. Conclusions
In conclusion, these findings demonstrate that Ap-HP6 serves as an essential component of the immune-associated serine protease cascade in A. pernyi hemolymph. It plays a pivotal role in pathogen-mediated PPO activation and contributes to the induction of AMP synthesis. Furthermore, in vitro hydrolysis of rAp-proHP6-Tb-His6 yielded active rAp-HP6 exhibiting serine protease activity, with optimal reaction conditions for substrate S-2288 hydrolysis identified as pH 8.0 at 50 °C for 15 min.”

Reviewer 2 Report
Comments and Suggestions for Authors
Authors address the reviewer's comments in this revised manuscript.
Author Response
Comments 1: Authors address the reviewer's comments in this revised manuscript.
Response 1: We are extremely grateful for the constructive comments and affirmations from the reviewer, which have greatly enhanced the scientific rigor and clarity of this work.
